# TCR-T Cell Recognition of an NY-ESO-1 Epitope Presented by HLA-A2 Supertype: Implications for Cancer Immunotherapy

**DOI:** 10.3390/vaccines13090898

**Published:** 2025-08-25

**Authors:** Qingqing Lin, Fenglan Liu, Yipeng Ma, Yanwei Li, Tong Lin, Xiaochun Chen, Jinling Zhang, Heng Sun, Zhi Wang, Xiaojun Xia, Geng Tian, Shi Jin, Mingjun Wang

**Affiliations:** 1Shenzhen Innovation Immunotechnology Co., Ltd., Shenzhen International Biological Valley-Life Science Industrial Park, Dapeng New District, Shenzhen 518119, China; qingqingl@szinno.com (Q.L.); fenglanl@szinno.com (F.L.); yipengma@hotmail.com (Y.M.); yanweil@szinno.com (Y.L.); lint@mail.sustech.edu.cn (T.L.); xiaochunc@szinno.com (X.C.); jinlingz@szinno.com (J.Z.); zhiwang@szinno.com (Z.W.); 2Department of Research and Development, Shenzhen Institute for Innovation and Translational Medicine, Shenzhen International Biological Valley-Life Science Industrial Park, Dapeng New District, Shenzhen 518119, China; 3National Cancer Center/National Clinical Research Center for Cancer/Cancer Hospital & Shenzhen Hospital, Chinese Academy of Medical Sciences and Peking Union Medical College, Shenzhen 518116, China; sunheng@cicams-sz.org.cn; 4State Key Laboratory of Oncology in South China, Guangdong Provincial Clinical Research Center for Cancer, Sun Yat-sen University Cancer Center, Guangzhou 510060, China; xiaxj@sysucc.org.cn; 5Laboratory of Cancer Immunology, Hainan Academy of Medical Sciences, Hainan Medical University, Haikou 571199, China; 6Department of Oncology, Shenzhen Second People’s Hospital, The First Affiliated Hospital of Shenzhen University, Shenzhen 518035, China; tiangeng666@aliyun.com; 7Biotherapy Centre, South China Hospital of Shenzhen University, Shenzhen 518111, China

**Keywords:** cancer immunotherapy, TCR-T, HLA-A2 alleles, HLA-A2 supertype, NY-ESO-1

## Abstract

Background: T-cell receptor (TCR)-engineered T-cell therapy (TCR-T) has become a promising anticancer therapy. Recognition of tumor cells by TCR-T cells requires matched human leukocyte antigen (HLA) alleles and tumor antigens, which seriously limits their population coverage. One strategy to expand the population coverage of a specific TCR-T cell therapy is to enable TCR-T cells to recognize target peptides presented by more HLA alleles. Methods: In this study, HLA alleles were selected based on the Chinese population frequency and HLA supertype classification. Then, COS-7 and two tumor cell lines (586 mel and 5637) were transduced with selected HLA alleles for functional evaluation of TCR-T cells. HLA-A2 alleles capable of both exogenously and endogenously presenting the NY-ESO-1-derived epitope and thereby being recognized by TCR-T cells were tested. Results: We demonstrated that a given TCR-T cell product can recognize the NY-ESO-1 peptide exogenously and endogenously presented not only by HLA-A*02:01 but also by HLA-A*02:03, HLA-A*02:06, and HLA-A*02:10, almost doubling the population coverage in the Chinese population from 12.01% to 21.05%. Conclusions: Our study suggests that cancer patients expressing members of the HLA-A2 supertype may benefit from the TCR-T cell product, and other TCR-T cell products could similarly expand their population coverage even within the non-Chinese population through an analogous approach.

## 1. Introduction

Although considerable advances have been made in the development of targeted therapy and immunotherapy, cancers are still one of the leading causes of death in most countries [1]. Therapies targeting activated driver oncogenes, such as EGFR, KRAS, and ALK, have brought cancer treatment to the era of precision therapy, but most patients eventually become drug-resistant due to tumor heterogeneity [2]. In recent years, immunotherapy has emerged as an alternative treatment for cancers. Immune checkpoint inhibitors targeting PD-1/PD-L1 and CTLA-4 have demonstrated durable clinical responses. However, only a small subset of cancer patients are sensitive to these treatments [3]. Chimeric antigen receptor T (CAR-T) cell therapy, a breakthrough in hematological cancers, has poor clinical efficacy in solid tumors [4]. T-cell receptor (TCR)-engineered T-cell therapy (TCR-T) has become an increasingly promising anticancer therapy after demonstrating good clinical efficacy in the treatment of metastatic melanoma, synovial sarcoma, non-small cell lung cancer, and human papillomavirus (HPV)-associated carcinomas [5,6,7,8]. On 1 August 2024, Adaptimmune (Abingdon, United Kingdom) received US FDA accelerated approval of TECELRA^®^ (afamitresgene autoleucel), the first approved TCR-T cell product for solid tumors, which confirmed the clinical potency of TCR-T cell therapy [9].

The recognition of target cells by TCR-T cells requires matching of both human leukocyte antigen (HLA) alleles and targeted proteins [10]. On the one hand, this mechanism guarantees the excellent precision of TCR-T cell therapy, which is the basis of its clinical safety and efficacy. However, this mechanism limits the population coverage of TCR-T cell therapy. For example, many TCR-T cell therapies currently under development are limited by HLA-A*02:01, which covers approximately 20–30% of the population in Western countries [11]. However, HLA-A*02:01 is only present in 12% of the Chinese population [12]. Considering that HLA alleles are among the most polymorphic genes, with more than 21,000 class I and 11,000 class II alleles [13], it is impossible to develop different TCRs targeting peptides presented by each HLA allele. One strategy to expand the population coverage of a specific TCR-T cell therapy would be to enable a given TCR-T cell product to recognize the target peptide presented by more HLA alleles. Based on the structural similarity of the epitope binding pocket, different HLA alleles can be classified as the HLA supertypes, which means that different HLA alleles from the same HLA supertype may bind to the same epitope, laying the foundation for recognition by a single TCR [14]. In 1999, Sette A. et al. established nine HLA supertypes (HLA-A1, HLA-A2, HLA-A3, HLA-A24, HLA-B7, HLA-B27, HLA-B44, HLA-B58, and HLA-B62) covering the majority of HLA-A and HLA-B polymorphisms based on endogenous binding peptide data and simple structural analysis [15]. Subsequently, the team further confirmed that over 80% of HLA-A and HLA-B alleles could be classified into nine previously defined HLA supertypes using newly accumulated HLA-I binding data and sequence information [14], validating the feasibility of the HLA supertype concept. Additionally, the Ole Lund team identified three new HLA-I supertypes (HLA-A26, HLA-B8, and HLA-B39) beyond the nine defined by the Sette’s group, based on their analysis of approximately 100 HLA-I-peptide interactions [16,17]. In 2023, another research team established a novel HLA supertype classification system based on significant correlations between the structural similarities of HLA-I molecules and peptide-binding specificities [18], providing a new direction for HLA supertype research. In addition to theoretical analysis, the HLA supertype concept is also supported by the fact that the newly approved TECELRA^®^ can recognize the MAGE-A4 peptide presented by a few HLA alleles (HLA-A*02:01, HLA-A*02:02, HLA-A*02:03, and HLA-A*02:06) [9].

Several approaches have been employed to evaluate the function of different HLA alleles in TCR-T cell therapy. One of them utilizes lymphoblastoid cell lines (LCLs) that naturally express different HLA alleles as antigen-presenting cells [19,20,21,22]. Although LCLs can be easily immortalized and cultured, it is difficult to collect all the required alleles, particularly those that are comparatively rare in the population. Moreover, the co-stimulatory molecules (e.g., CD80 and CD86) of LCLs can contribute to the activation of TCR-T cells in vitro, which cannot mimic real interactions with tumor cells. Moreover, the corresponding epitope was loaded onto LCLs by exogenous incubation with peptides, and the endogenous antigen processing process was not evaluated. Another viable approach is to directly employ tumor cell lines with different HLA alleles. However, similar to LCLs, collecting tumor cells that match both HLA alleles and tumor antigens is difficult. Moreover, the surface expression of HLA molecules can be impaired by the immune evasion strategies of tumor cells [23]. Therefore, studies employing this approach are limited. In addition, T2 cell-based HLA monospecific cells loaded with HLA peptide monomers have also been developed and can be used for the evaluation of T cell function across different HLA allotypes [24]. Similar to LCLs, this approach lacks the ability to evaluate intracellular antigen processing. Finally, computer-based predictions are included. Although the reliability of computer-aided predictions of T cell epitopes has increased tremendously in recent years [25,26], the subsequent functional recognition of HLA-peptide complexes by TCRs still requires verification by wet laboratory experiments.

In the present study, by using an HLA-A*02:01-restricted NY-ESO-1-specific TCR, we developed a novel strategy for expanding the population coverage of TCR-T cells. HLA alleles were selected based on their population frequency and HLA supertype classification [16], which can be a practical approach for selecting a reasonable number of potentially reactive HLA alleles for functional evaluation. Then, COS-7 and two tumor cell lines (586 mel and 5637) were used for the transduction of selected HLA alleles for functional evaluation of TCR-T cells. Finally, we demonstrated that a given TCR-T cell product could recognize the NY-ESO-1 peptide presented not only by HLA-A*02:01 but also by HLA-A*02:03, HLA-A*02:06, and HLA-A*02:10.

## 2. Materials and Methods

### 2.1. Cells

293T and T2 cells were purchased from the American Type Culture Collection (ATCC). 5637 and COS-7 cells were purchased from the Cell Bank of the Chinese Academy of Sciences. In total, 586 mel cells were kindly provided by Dr. Rongfu Wang from the University of Southern California. NY-ESO-1 TCR-T cells, called 1G4-α95:LY TCR-T cells, were obtained as described previously [7,8]. 1G4-α95:LY TCR contained two amino acid substitutions in the third complementarity-determining region of the native 1G4 TCR α chain to enhance its ability to recognize target cells [8,27]. 293T, 586 mel, 5637, and COS-7 cells were kept in-house and cultured in DMEM (Corning) supplemented with 10% fetal bovine serum (FBS, LONZA), 1% penicillin-streptomycin, and 1% GlutaMAX supplement (Gibco).

### 2.2. Generation of COS-7-A2, COS-7-NY-A2, 586 mel-A2, and 5637-NY-A2 Cells

Nucleotide sequences of NY-ESO-1, HLA-A*02:01, HLA-A*02:03, HLA-A*02:05, HLA-A*02:06, HLA-A*02:07, HLA-A*02:09, HLA-A*02:10, HLA-A*02:11, and HLA-A*02:48 were extracted from the IMGT database [28]. They were synthesized by Sangon Biotech (Shanghai, China) and integrated into the pMSGV1 vector [29]. 293T cells were co-transfected with pMSGV1, VSV-G, and gag-pol plasmids to generate the viruses. COS-7 and 586 mel (HLA-A2-, NY-ESO-1+) cells were infected with the virus containing different HLA-A2 alleles individually. The expression of HLA-A2 (except for HLA-A*02:10) was verified using flow cytometry and RT-PCR. The expression of HLA-A*02:10 was verified using RT-PCR alone. COS-7-A2 and 586 mel-A2 cells were enriched using anti-PE microbeads from Miltenyi Biotec, Köln, Germany (130-048-801). To generate COS-7-NY-A2 and 5637-NY cells, genes encoding NY-ESO-1 were transduced into COS-7-A2 and 5637 cells, and for the purpose of detection, genes for Green Fluorescent Protein (GFP) were simultaneously introduced. Subsequently, HLA-A*02:01, HLA-A*02:03, HLA-A*02:05, HLA-A*02:06, and HLA-A*02:10 were introduced into 5637-NY cells, generating 5637-NY-A2 cells. The expression of HLA-A2 in these cells was verified using a previously described method.

### 2.3. Incubation of the TCR-T Cells with COS-7-A2, COS-7-NY-A2, 586 mel-A2, and 5637-NY-A2 Cells

TCR-T cells were incubated with COS-7-A2 cells stably transfected with HLA-A*02:01, HLA-A*02:03, HLA-A*02:05, HLA-A*02:06, HLA-A*02:07, HLA-A*02:09, HLA-A*02:10, HLA-A*02:11, and HLA-A*02:48, with the wild type (WT) as a non-transfected control. COS-7-A2 cells were loaded with 10 μM of the corresponding peptide for 2 h at 37 °C and then washed twice prior to co-culture. After 16–24 h, supernatants were collected and analyzed for IFN-γ by ELISA. COS-7-NY-A2, 586 mel-A2, 586 mel-WT, 5637-NY-A2, 5637-NY, 5637-WT, or 624 mel cells were co-cultured with TCR-T cells in the same manner.

### 2.4. ELISA

An anti-IFN-γ monoclonal antibody was used as the primary antibody (Clone 2G1, Invitrogen, M700A, Carlsbad, CA, USA), which was coated onto a MaxiSorp 96-well plate (Nunc, 442404) overnight at 4 °C. The antibody was removed, and the plates were blocked with PBS supplemented with 1% BSA for 2 h. The cells were co-cultured as previously described, and the supernatants were transferred to precoated plates for incubation. A biotin-labeled anti-IFN-γ antibody was used as the secondary antibody (clone B133.5, M701b, Invitrogen), and HRP-conjugated streptavidin (Thermo Scientific, N200, Waltham, MA, USA) was added after secondary antibody incubation. Then, 100 μL/well TMB (Sigma Aldrich, St. Louis, MO, USA, T5525) was added as the substrate for HRP. The plates were read using an Infinite M200 PRO reader (TECAN Austria GmbH, Grödlg, Austria).

### 2.5. Flow Cytometry (FACS) Analysis

T cells were analyzed by flow cytometry (FCM). Briefly, the cells were suspended in FCM buffer containing sterile phosphate-buffered saline (PBS, Corning, 21-040-CV) supplemented with 1% bovine serum albumin (Roche, 10735078001). Transduction efficiency of HLA-A2 alleles was assessed 48 h after transduction by staining for HLA-A2 (clone BB7.2, BD Biosciences, 558570). 7-AAD (559925; BD Biosciences) was used to remove dead cells. For TCR-T cells, the following markers were used: PE-eF610-CD3 (clone UCHT1, Invitrogen, 61-0038-42), TCR Vβ 13.1-APC (clone H131, Biolegend, 362408), APC-H7-CD8 (clone SK1, BD Biosciences, 561423), Alexa Fluor 700-CD4 (clone SK3, BD Biosciences, 566318), BV421-4-1BB (clone 4B4-1, BD Biosciences, 564091), PE-PD-1 (clone EH12.1, BD Biosciences, 560795), and BV785-LAG-3 (clone 11C3C65, Biolegend, 369322). The stained cells were washed and resuspended in FCM buffer and analyzed using a FACS Aria II (BD Biosciences, San Jose, CA, USA). 

### 2.6. Statistical Analysis

Statistical analyses were performed using GraphPad Prism version 6.01 for Windows (GraphPad Software, San Diego, CA, USA). For multiple comparisons between groups, we employed multiple *t*-tests—one per row with a Holm-Sidak correction to assess statistical significance. The significance threshold was set at *p* < 0.05. For correlations between groups, we employed Pearson’s r test of correlation, and *p* < 0.05 was considered to indicate a statistically significant difference. Data are presented as mean ± standard deviation (mean ± SD). EC50 values were calculated using nonlinear regression analysis, specifically a four-parameter logistic model, to fit the concentration–response curves.

## 3. Results

### 3.1. Population Frequency-Based Selection of HLA Supertype Genes for the Expansion of Population Coverage of a Given TCR

A selection strategy combining the population frequency of HLA subtypes and HLA supertype classifications was employed. As shown in Appendix A, the top 10 HLA-A2 supertype alleles were chosen based on their frequencies in the Chinese population and HLA supertype classification [12,14]. The fact that the total frequency of the selected alleles reaches 99.85% and the least frequent allele of HLA-A*02:48 has a low frequency of 0.0097% in the population indicates that these 10 selected alleles can cover most HLA-A2 supertype genes in the population. Moreover, given that HLA alleles are among the most polymorphic genes, HLA-A2 alleles were functionally clustered based on predicted binding motifs using MHCcluster (Appendix A) [30]. Although the 10 alleles were all classified as the HLA-A2 supertype, functional differences still existed. While HLA-A*02:01, HLA-A*02:09, and HLA-A*02:11 are closely related, HLA-A*02:06, HLA-A*02:10, HLA-A*02:05, and HLA-A*02:03 exhibit distinct differences. To evaluate the functional spectrum of TCR-T cell therapy, 10 HLA-A2 alleles were transduced into COS-7 cells and 586 mel cells (NY-ESO-1+ HLA-A2−). The expression of HLA-A2 subtypes was verified using FACS and/or RT-PCR (Figure 1a,b, Appendix A). COS-7-NY-A2 and 5637-NY-A2 cells were also constructed to evaluate the ability of HLA superfamily members to present endogenously processed antigen epitopes, and the expression of HLA-A2 and NY-ESO-1 in these cells is shown in Appendix A. Because HLA-A*02:53N was a truncated allele, meaning that it did not have normal HLA function, it was excluded from this and the following analysis.

### 3.2. Target Antigen Peptide Presented by Members of the HLA-A2 Supertype in Cos-7 Cells Could Activate the HLA-A*02:01-Restricted TCR-T Cells

To investigate the potential presentation of the target antigen peptide by different HLA-A2 molecules, COS-7 wild-type (WT) and COS-7 cells transduced with different HLA-A2 subtype genes (COS-7-A2) were loaded with the corresponding NY-ESO-1 157-165 epitope (SLLMWITQC) and then co-incubated with 1G4-α95:LY TCR-T cells derived from HLA-A*02:01+ patients [27]. As shown in Figure 2a, TCR-T cells not only recognized the NY-ESO-1 peptide presented by HLA-A*02:01 but also HLA-A*02:03, HLA-A*02:05, HLA-A*02:06, and HLA-A*02:10. Limited activity was observed for the NY-ESO-1 peptide with HLA-A*02:09 and HLA-A*02:11 alleles. TCR-T cells were not active for the peptides presented by HLA-A*02:07 and HLA-A*02:48. As negative controls, no activity was found for COS-7 WT or COS-7-A2 cells loaded with an epitope from the influenza virus (Flu 58–66). Moreover, all COS-7-A2 control T-cells were inactive (Appendix A). 

After preliminary selection of HLA-A2 supertype candidates by loading the exogenous tumor antigen epitope onto COS-7-A2 cells, NY-ESO-1 genes were further transduced into COS-7-A2 cells (COS-7-NY-A2) to evaluate the ability of HLA superfamily members to present endogenously processed antigen epitopes. As depicted in Figure 2b, endogenous NY-ESO-1 peptide presented by HLA-A*02:01, HLA-A*02:03, HLA-A*02:05, HLA-A*02:06, and HLA-A*02:10 significantly induced TCR-T cells to secrete higher levels of IFN-γ than control T cells. The responses elicited by the peptide in complex with HLA-A*02:05 and HLA-A*02:10 were substantially lower than those induced by the other HLA-A2 alleles.

### 3.3. Evaluation of the Functional Avidity of Target Antigen Peptide Presented by Different HLA-A2 Alleles of the HLA-A2 Supertype

To quantify the functional avidity of TCR-T cells for the five active HLA-A2 alleles (A*02:01, A*02:03, A*02:05, A*02:06, and A*02:10), the antigen peptide (SLLMWITQC) was loaded onto COS-7-A2 and T2 cells in serial dilutions. As shown in Figure 3a–f, TCR-T cells exhibited concentration-dependent recognition of different HLA-A2 alleles. In general, TCR-T cells showed the greatest functional avidity for T2 cells, with an EC50 of 0.004 μM. The functional avidity of COS-7-A*02:01 was lower than that of the T2 cells, with an EC50 of 0.036 μM. The EC50 values of functional avidity for A*02:03, A*02:05, A*02:06, and A*02:10 were 0.995 μM, 0.188 μM, 0.351 μM, and 0.239 μM, respectively.

### 3.4. Target Antigen Peptide Could Be Presented by Transduced HLA-A2 Supertype Members in Tumor Cells and Be Recognized by HLA-A*02:01-Restricted TCR-T Cells

The presentation of endogenous epitopes in tumor cells is a complex process involving proteolysis, peptide transportation, peptide binding to HLA alleles, etc. To evaluate this process, 586 mel (NY-ESO-1+A2-) was used for the transduction of different HLA-A2 alleles. Similar to COS-7-A2 cells, TCR-T cells also exhibited activity for different HLA-A2 alleles other than HLA-A*02:01 according to ELISA (Figure 4a). In general, the activities were most potent for HLA-A*02:01, HLA-A*02:10, HLA-A*02:03, and HLA-A*02:06 but not for HLA-A*02:05, which is different from the results obtained for COS-7-A2 cells. Furthermore, 624 mel cells, which are naturally double-positive cells (HLA-A*02:01+, NY-ESO-1+), exhibited potent activity toward TCR-T cells. As expected, no activity was detected for the 586 mel WT strain. The activation markers (4-1BB, PD-1, and LAG-3) of the T cells were further evaluated via FACS analysis. As shown in Figure 4b and 4c, the expression of 4-1BB increased significantly after incubation with 586 mel-A*02:01, 586 mel-A*02:10, 586 mel-A*02:03, 586 mel-A*02:06, and 624 mel. The expression of LAG-3 and PD-1 only showed minor changes. A strong positive correlation between IFN-γ secretion, as determined by ELISA, and 4-1BB upregulation in FACS analysis was identified (Figure 4d). 

In addition, similar to 586 mel-A2 cells, TCR-T cells exhibited activity for different HLA-A2 alleles other than HLA-A*02:01 in 5637-NY-A2 cells (Appendix A). As expected, no activity was detected in the 586 mel-WT, 5637-WT, and 5637-NY strains. 

## 4. Discussion

Despite being a promising precision medicine for tumor treatment, the limited population coverage of TCR-T cell therapy hinders its clinical application. Each HLA allele can potentially possess a unique repertoire of binding peptides, considering that most HLA allele polymorphisms are located in the epitope-binding region. However, despite this polymorphism, HLA class I molecules can be clustered into supertypes that share overlapping peptide binding specificity and present a given epitope to be recognized by a single TCR [31,32]. A novel strategy for expanding the population coverage of TCR T-cells was developed and evaluated in this study.

The top 10 HLA-A2 supertype alleles were selected based on their population frequencies in the Chinese population. The 10 chosen alleles can cover 99.85% of the entire HLA-A2 population with the potential to be recognized by the HLA-A*02:01-restricted TCR, considering that they may bind with the same NY-ESO-1 157-165 epitope. Among them, HLA-A*02:48 has a frequency of 0.0097% and is a rare allele encountered in the clinical use of TCR-T cell therapy. These findings suggest that the selected alleles encompass nearly all clinically relevant HLA-A2 supertype variants. The functional clustering of these 10 alleles suggested that there are certain differences among the peptide-binding capacities of different alleles, although they belong to the same supertype. Furthermore, it is well known that the amino acid substitutions that define the different HLA-A2 subtypes can potentially influence not only peptide binding but also the conformation of the peptide/HLA complex and thus its recognition by the TCR, which also warrants further functional evaluation of TCR-T cells harboring different supertype alleles.

One advantage of using an exogenously loaded peptide is that a particular epitope from an antigen can be clearly targeted and evaluated. In this study, the specific recognition of the NY-ESO-1 157-165 epitope (SLLMWITQC) by TCR-T cells over the control peptide loaded on COS-7-A2 cells suggests that this TCR specifically recognizes the NY-ESO-1 157-165 epitope. A similar level of IFN-γ release by TCR-T cells activated by either peptide-loaded COS-7-A*02:01 or T2 cells confirms the reliability of using transgenic HLA-A2 alleles. The highest activity of TCR-T cells activated by the NY-ESO-1 157-165 loaded on HLA-A*02:01-expressing antigen-presenting cells (APCs) is expected because it is an endogenous TCR identified in HLA-A*02:01 patients. Interestingly, for the first time, we found that this TCR is also reactive to NY-ESO-1 157-165-loaded APC expressing HLA-A*02:03, HLA-A*02:05, HLA-A*02:06, or HLA-A*02:10, in addition to HLA-A*02:01. In addition, we introduced a number of HLA-I alleles (such as HLA-A*11:01 and HLA-A*24:02) that were highly frequent in the Chinese population into COS-7 cells, and only the antigen peptide presented by COS-7 transfected with HLA-A*02:01 could stimulate TCR-T cells to release high levels of IFN-γ, indicating that the TCR-T response is indeed HLA-A2-restricted (Appendix A), and no off-target toxicity was observed between the similar human peptides and TCR-T cells (Appendix A). In addition, the endogenously processed antigen epitopes presented by HLA-A2 alleles can also be recognized by TCR-T cells. Therefore, this finding may expand the population coverage of TCR-T cells from 12.01% to 21.05%. Interestingly, the four reactive HLA-A2 alleles showed less functional similarity with HLA-A*02:01 than the nonreactive alleles in the clustering of functional relationships (Appendix A) [30]. This suggests that the polymorphism located outside the binding pocket has tremendous effects on TCR recognition, which highlights the need for functional evaluation instead of peptide-HLA binding tests. In addition, activation of TCR-T cells induced by COS-7-NY-A2 cells further suggested that the five HLA-A2 alleles could present exogenous as well as endogenous NY-ESO-1 epitopes.

To investigate why TCR-T cells do not respond to the epitope presented by other members of the HLA-A2 supertype, such as HLA-A*02:07, HLA-A*02:48, HLA-A*02:09, and HLA-A*02:11, we compared the sequences of HLA-A2 alleles (Appendix A). The HLA-A protein regions were obtained from UniProt (https://www.uniprot.org/) and are listed in Appendix A. The MHC-I binding prediction results by NetMHCpan 4.1 [33] are listed in Appendix A. Recognition of the peptide-major histocompatibility complex (pMHC) by TCR-T cells may be influenced by three factors: the binding affinity of antigen epitopes to HLA alleles, the affinity of pMHC for TCR, and the stability of the pMHC complex. Alpha-1 and Alpha-2 regions are peptide-binding clefts that directly affect the affinity of HLA-A2 alleles and target antigens. In these two regions, the substitution of R for G at AA Pos. 65 (HLA-A*02:48), T for I at AA Pos. 73 (HLA-A*02:11), and Y for C at Pos. 99 (HLA-A*02:07) prevented TCR-T cells from recognizing the corresponding pMHCs. However, the predicted binding affinities of these HLA-A2 alleles were the same or even lower than those of selected HLA-A2 superfamily members. Considering that peptides at a high concentration of 10 μM were added 2 h before co-incubation, TCR-T cells could immediately encounter target cells. Thus, the lower binding affinity of pMHC with TCR caused by sequence changes might be the key reason, as the stability of pMHC is unlikely to play a role here. For HLA-A*02:09, the substitution of A to E in the Alpha-3 region may affect the interaction of pMHC with the CD8 co-receptor, which further influences the recognition of TCR-T cells.

Additionally, we assessed the functional avidity of TCR-T cells against the NY-ESO-1 peptide presented by HLA-A2 supertype alleles by EC50, defined as the peptide concentration inducing half-maximal activation of the T-cell population. Theoretically, low EC50 values indicate that the TCR can recognize tumor cells with low epitope density, potentially translating to improved clinical efficacy. For instance, a study optimizing an MAGE-A1-specific TCR via somatic hypermutation demonstrated that T cells with enhanced avidity exhibited significantly higher IFN-γ production and cytotoxicity against melanoma cells, both in vitro and in mouse models [34]. In a clinical trial by Steven A. Rosenberg and colleagues, patients treated with TCR-T cells bearing low-avidity (DMF4) versus high-avidity (DMF5) TCRs targeting the same MART-1 antigen achieved objective tumor response rates (ORRs) of 13% (4 of 34) and 30% (6 of 20), respectively, suggesting that increased TCR avidity may enhance tumor rejection [35]. In our study, we observed a range of EC50 values across alleles (0.036 to 0.995 μM), a difference that reasonably raises questions about the clinical efficacy of lower-avidity alleles (e.g., HLA-A*02:03, A02:06, and A*02:10). However, multiple lines of evidence mitigate these concerns: First, clinical studies support that TCR avidity is not the sole determinant of therapeutic outcome. Steven A. Rosenberg and colleagues noted in their trials that TCR avidity may not be a major driver of tumor rejection, emphasizing that other factors (e.g., T cell persistence and tumor microenvironment) also play critical roles [35,36]. Second, our in vitro data confirm functional activation of TCR-T cells by these lower-avidity alleles. In tumor cell co-cultures, HLA-A*02:03+, A*02:06+, and A*02:10+ cells induced IFN-γ secretion and 4-1BB upregulation at levels comparable to HLA-A*02:01 (Figure 4a–c), indicating robust TCR engagement despite higher EC50 values. Third, clinical evidence directly validates this functionality. In our trial (NCT02457650), an HLA-A*02:03+ patient, whose allele has a higher EC50 (0.995 μM vs. 0.036 μM for HLA-A*02:01), achieved a partial response, demonstrating that lower avidity does not preclude meaningful clinical benefit [7]. Finally, it has been shown that T cell activation and signaling are enhanced up to a specific TCR-pMHC affinity threshold, beyond which T cells may fail to develop productive functions [36]. Thus, while EC50 is an important parameter, it is not a definitive predictor of TCR-mediated cytotoxic efficacy or therapeutic index. 

In addition, we compared the avidity obtained by co-culturing TCR-T cells with COS-7-A2 cells pulsed with antigen epitopes. A strong positive correlation was found between avidity and the predicted binding affinity of the antigen epitope for HLA-A*02:01, HLA-A*02:03, HLA-A*02:06, and HLA-A*02:10 (Appendix A). However, when HLA-A*02:05 was included in the analysis, a weak correlation was observed (Appendix A). The reason might be that avidity reflected the recognition of TCR for pMHC, but the predicted binding affinity only indicated the ability of the antigen epitope to bind with HLA-A2 alleles, which was just a factor that might affect avidity. 

One of the disadvantages of both COS-7-A2 cells and commonly used LCLs is that the evaluation of antigen processing is not considered during functional tests of TCR-T cells. This process can have tremendous effects on the surface presentation of epitopes, because the alleles are required to bind with the processed short peptide before being transported to the cell surface. Therefore, the inclusion of antigen-positive COS-7-NY-A2 and tumor cells is critical for overcoming these drawbacks. NY-ESO-1 can be detected in multiple tumor cells; therefore, two types of tumor cells were chosen to verify the members of the HLA-A2 supertype. Surprisingly, HLA-A*02:05 in antigen-presenting cells could present endogenously processed peptides to TCR-T cells within the context of COS-7 cells but failed to do so within tumor cells. However, when tumor-A*02:05 cells were loaded with the exogenous NY-ESO-1 epitope, TCR-T cells could still recognize them. Furthermore, a strong positive correlation between IFN-γ secretion and 4-1BB upregulation was demonstrated in TCR-T cells upon co-culture with 586 mel-A2 cells, with only minor fluctuations in the expression of LAG-3 and PD-1. To further validate these conclusions, we will perform experiments using HLA-A2+ NY-ESO-1+ patient-derived organoids or primary cells in future studies, including mass spectrometry (MS) analysis and co-culture assays with TCR-T cells.

Differences in cell types may be one reason for this finding. COS-7 cells, a fibroblast-like cell line derived from African green monkey kidney, function as “non-professional” antigen-presenting cells (APCs) and lack MHC class II and costimulatory molecules [37,38]. Studies have demonstrated that, similar to human cells, transporter associated with antigen processing (TAP) in COS cells can be inhibited by the TAP inhibitor ICP47, indicating similarities between human and monkey TAP [39,40]. They have been widely used to study T cell function in response to peptide-MHC (pMHC) complexes through transfection of HLA alleles and oncogenes [41,42]. Additionally, FACS results (Appendix A) and ELISA data (Figure 2b) confirmed that the NY-ESO-1 protein was successfully expressed and presented on the surface of COS-7 cells in association with HLA-A2 alleles. However, deficiencies in antigen processing machinery have been reported as an important mechanism for immune escape in tumor cells. For 586 mel cells, the loss of HLA-A29, HLA-B44, and HLA-DR7 alleles, all part of the original patient’s haplotype (A29,31; B8,44; and DR1,7), was reported to result from a complete loss of a genomic unit [43,44]. Additionally, downregulation of interferon-stimulated genes (ISGs), including STAT1, TMEM173, and OAS3, has been detected in 5637 cells [45]. Specifically, diminished STAT1 activation can lead to low HLA class I expression, which impairs the effectiveness of cytotoxic T lymphocyte (CTL) responses in mediating tumor elimination and mediates TAP1-dependent escape from CTL [45,46]. Therefore, COS-7 cells cannot perfectly mimic the antigen processing and presentation processes in tumor cells.

Additionally, the stability of peptide-MHC (pMHC) complexes on target cells may represent another contributing factor. The affinity of peptide binding to MHC-I is important for the formation of pMHC, and multiple prediction methods have been developed, such as NetMHC 4.1 and MHCflurry [47]. However, some studies have revealed that binding affinity is not the only determinant of peptide immunogenicity [48]. pMHC should not only be formed but also retained on the cell surface of target cells before encountering the corresponding T cells. Therefore, it has been suggested that the stability of pMHC is a better predictor of peptide immunogenicity than its affinity [49,50,51]. Harndahl et al. revealed that the predicted half-lives of approximately 30% of non-immunogenic peptides are lower than 1 h [52]. Some tools have been developed to predict binding stability, such as NetMHCstab [53], NetMHCstabpan 1.0 [54], TLStab, and TLImm [55]. As shown in Appendix A, the limited presentation capacity of HLA-A*02:05 in tumor cells may reflect its low predicted affinity (NetMHCpan %Rank = 3.946) and complex stability (NetMHCstabpan %Rank = 4.00), which could impact endogenous peptide presentation efficiency despite exogenous rescue. Although relevant wet laboratory experiments (e.g., monitoring the dissociation of pMHC complexes at 37 °C) should be conducted in future studies to validate the accuracy of these predicted pMHC stability results, this could partly explain why HLA-A*02:05 fails to present endogenously processed peptides—given that the corresponding pMHCs exhibit low binding affinity and stability. While high concentrations of exogenously loaded peptide might compensate, the formation of pMHC for endogenously processed epitopes in tumor-A*02:05 cells may be severely impaired. By combining affinity and stability data, we can predict members of the HLA supertype family more accurately. However, the prediction and experimental results of HLA-A*02:07, HLA-A*02:09, HLA-A*02:11, and HLA-A*02:48 suggest that the stability and binding affinity of pMHC are still not all determinants of TCR-T cell immune response, and further exploration remains to be performed. Moreover, this difference highlights the necessity of including tumor cell lines for the functional evaluation of TCR-T cells. 

As mentioned above, it is critical to include both exogenously and endogenously presented epitopes in the functional evaluation of TCR-T cells, and a strong positive correlation between these two different assays is required for the functional verification of TCR-T cells for different HLA alleles as well as the particular epitope (Appendix A). However, with one deviation point for HLA-A*02:05, functional evaluation of COS-7-A2 and 586 mel-A2 cells revealed a strong correlation. This finding further confirms both the specific activity of this TCR for the NY-ESO-1 157-165 epitope and the reliability of this strategy for functionally evaluating different HLA alleles. The combined results of these assays expand the population coverage of the evaluated TCR-T cells for the treatment of patients expressing HLA-A*02:01, A*02:03, A*02:06, or A*02:10. The results of incubating TCR-T cells with two other tumor-NY-A2 cells further support this conclusion. For the first time, the population coverage of TCR-T cells expanded from 12.01% to 21.05%. In addition, for the non-Chinese population, the population coverage of TCR-T cells could also be increased. According to the data collected from the Allele Frequency Net Database (AFND) [11] shown in Appendix A, for Asian countries such as Japan and India, a significant increase in population coverage with the given HLA-A2 alleles was observed in Japan from 11.62% to 21.15% and in India from 4.67% to 7.56%. However, for non-Asian people, the enhancement would be only 0.09–0.18% for the United States and 0.19% for Germany. The difference in allele frequencies among populations indicates that, for a given TCR-T cell aiming for a non-Chinese population, different alleles should be chosen as candidates of HLA supertype according to their frequencies among the target population to achieve the best improvement. Our conclusion is also supported by the fact that the newly approved TECELRA^®^ can recognize the MAGE-A4 peptide presented by a few HLA alleles (HLA-A*02:01, HLA-A*02:02, HLA-A*02:03, and HLA-A*02:06) [9]. In addition, we have incorporated the HLA-A2 supertype concept into a clinical trial (NCT02457650) designed to evaluate the safety and feasibility of 1G4-α95:LY TCR-T cell therapy in HLA-A2+ NY-ESO-1+ tumor patients. Four HLA-A2+ non-small cell lung cancer (NSCLC) patients were enrolled, with no serious adverse events observed post-infusion [7]. Two patients achieved clinical responses: one HLA-A*02:03+ patient had a partial response (PR), and another HLA-A*02:01+ patient achieved stable disease (SD). These findings support the clinical potential of the HLA-A2 supertype concept.

Nonetheless, we acknowledge the need for further validation to strengthen these conclusions. Our prospective research agenda includes two key components: First, establishing humanized xenograft models in NSG mice, where tumor cells engineered to express HLA-A2 supertype alleles will be implanted, followed by adoptive transfer of 1G4-α95:LY TCR-T cells to assess allele-specific tumor rejection and persistence. Second, expanding the clinical trial to enroll more patients across these HLA-A2 subtypes, with monitoring of objective response rates and correlation with TCR-T cell expansion and function in vivo.

## 5. Conclusions

In summary, we demonstrated that HLA-A*02:01-restricted TCR-T cells might be used not only for patients expressing HLA-A*02:01 but also for patients expressing HLA-A*02:03, HLA-A*02:06, or HLA-A*02:10. Based on the frequency of these HLA alleles, the population coverage almost doubled from 12.01% to 21.05%. More importantly, the HLA selection strategy based on population frequencies and HLA supertype classification has proven viable for the evaluation and expansion of population coverage of TCR-T cell therapies. Our results suggest that artificial expression of HLA supertype genes in both COS-7 cells and antigen-positive cells is necessary for the functional verification of TCR-T cells for a particular epitope.

## Figures and Tables

**Figure 1 vaccines-13-00898-f001:**
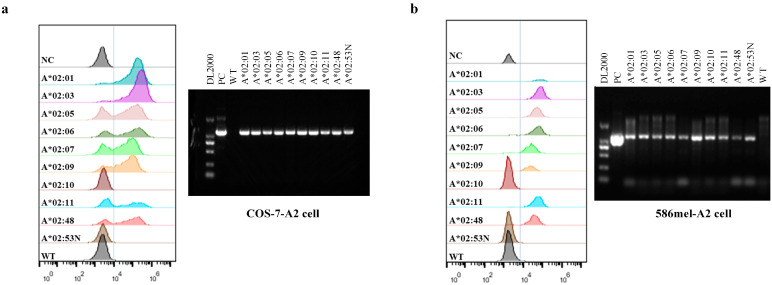
Transfection efficiency of transfectants. FACS and RT-PCR were performed to analyze HLA-A2 expression in COS-7-A2 cells (**a**) and 586 mel-A2 cells (**b**). Among all the selected HLA-A2 alleles, the expression of all except HLA-A*02:10 and HLA-A*02:53N were detectable by FACS. Specifically, HLA-A*02:10 and HLA-A*02:53N could not be detected using the FACS antibody; however, RT-PCR results confirmed the successful introduction and expression of these two alleles in the corresponding transfected cells.

**Figure 2 vaccines-13-00898-f002:**
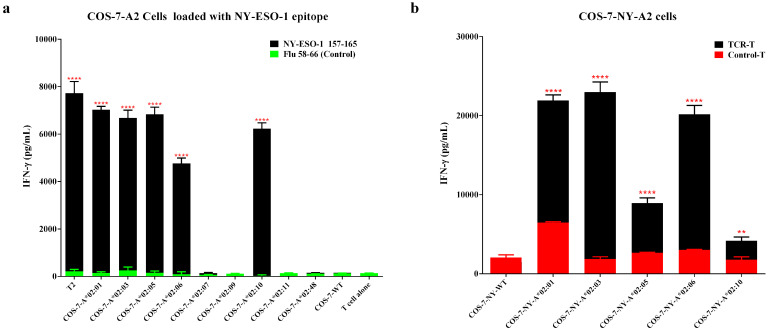
TCR-T cells restricted by HLA-A*02:01 demonstrated the ability to recognize the target antigen peptide presented by members of the HLA-A2 supertype in COS-7 cells. IFN-γ secretion results of incubating TCR-T cells with COS-7-A2 cells loaded with different peptides (**a**) or COS-7-NY-A2 cells (**b**). The experiment was performed in triplicate wells, and data are representative of three independent experiments (n = 3). Data are represented as mean ± SD. ** *p* < 0.01, and **** *p* < 0.0001.

**Figure 3 vaccines-13-00898-f003:**
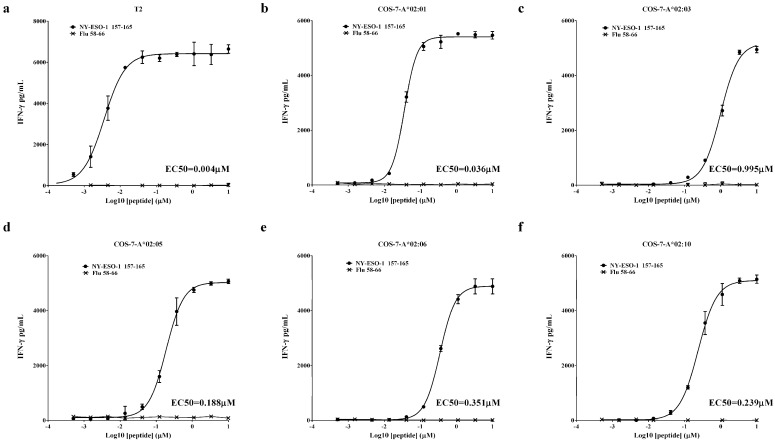
Functional avidity of target antigen peptide presented by members of the HLA-A2 supertype. TCR-T cells were co-cultured with T2 cells (**a**) and COS-7 cells expressing one of 5 active HLA-A2 alleles (A*02:01, A*02:03, A*02:05, A*02:06, and A*02:10) (**b**–**f**) loaded with antigen peptide (SLLMWITQC) at different concentrations. IFN-γ secretion was detected by ELISA, and EC50 was calculated using nonlinear regression analysis, specifically a four-parameter logistic model, to fit the concentration–response curves to evaluate functional avidity. The experiment was performed in triplicate wells, and data are representative of four independent experiments (n = 4). Data are represented as mean ± SD.

**Figure 4 vaccines-13-00898-f004:**
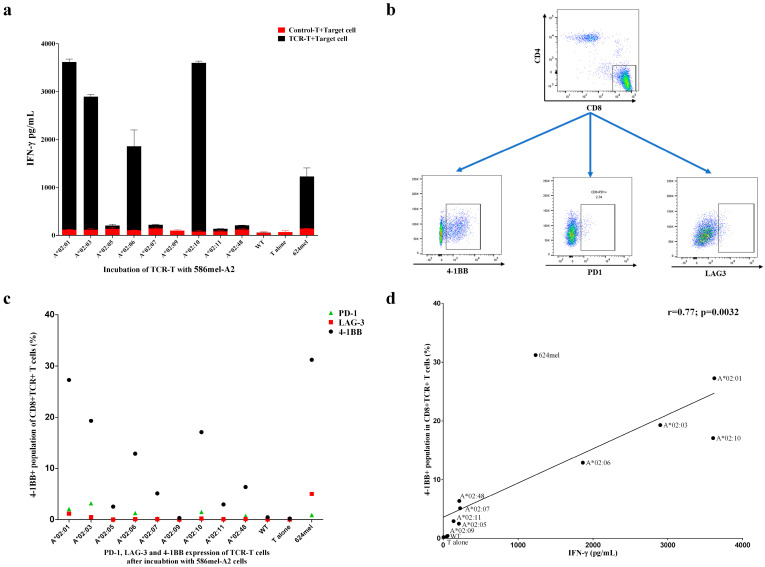
HLA-A*02:01-restricted TCR-T cells could recognize target antigen peptide presented by members of the HLA-A2 supertype in 586 mel cells. (**a**) IFN-γ secretion results of incubating TCR-T cells and control T cells with 586 mel-A2 cells, 586 mel WT cells, and 624 mel cells. The experiment was performed in triplicate wells, and data are representative of four independent experiments (n = 4). Data are represented as mean ± SD. The difference between TCR-T and Control-T cells was calculated by using multiple *t*-tests—one per row with GraphPad Prism 6.01 software. (**b**,**c**) Detection of activation markers on T cells by FACS. Data are representative of two independent experiments (n = 2). (**d**) Correlation between IFN-γ secretion and 4-1BB upregulation. The statistical analysis was performed using the Pearson r test of correlation with GraphPad Prism 6.01 software, and *p* < 0.05 was considered to indicate a statistically significant difference.

## Data Availability

The original contributions presented in this study are included in the article/Appendix A. Further inquiries can be directed to the corresponding authors.

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
