# Peer review of "TCR-T Cell Recognition of an NY-ESO-1 Epitope Presented by HLA-A2 Supertype: Implications for Cancer Immunotherapy"

_vaccines, 2025, doi:10.3390/vaccines13090898_

Round 1
Reviewer 1 Report
Comments and Suggestions for Authors
The manuscript presents a timely and methodologically solid study that proposes a population-based HLA supertype selection strategy to enhance the clinical applicability of TCR-T cell therapy targeting NY-ESO-1. The combination of functional assays using both COS-7 and tumor cell models strengthens the conclusions.
To improve clarity and reader understanding, I recommend the following:
-
Clarify TCR Origin and Specificity
Early in the Results section (e.g., Section 3.1, page 5), please state that the 1G4-α95:LY TCR used in the study is derived from HLA-A02:01+ patients. This context helps explain the high specificity observed for A02:01 in peptide recognition assays. -
Explain Functional Clustering
In Supplementary Figure 1b, the functional clustering of HLA-A2 alleles is an important analytical step. However, the main text (page 5) refers to it without explaining how it was computed. Adding a short sentence noting that clustering was done using MHCcluster [30] would improve transparency.Example: “As shown in Supplementary Figure 1b, HLA-A2 alleles were functionally clustered based on predicted binding motifs using MHCcluster [30].”
-
Discuss Discrepancies in Avidity – A*02:05
The Discussion (page 10) mentions that HLA-A*02:05 shows low avidity in tumor cells but not in COS-7 cells. This discrepancy should be tied more clearly to the predicted low peptide binding stability (NetMHCstabpan score = 4.00, Supplementary Table 5), and to the possibility that high exogenous peptide concentration in COS-7 experiments compensates for weak endogenous processing.Example: “The limited presentation capacity of HLA-A*02:05 in tumor cells may reflect its low predicted complex stability (NetMHCstabpan %Rank = 4.00), which could impact endogenous peptide presentation efficiency despite exogenous rescue.”
-
Language Polishing
The English is mostly clear but can be improved in a few places:-
Page 5: “Minor activity was observed...” → “Limited activity was observed...”
-
Page 10: “...warrants the need for functional evaluation...” → “...highlights the need for functional evaluation...”
-
-
Future Directions
In the Conclusions (page 11), you mention follow-up studies in animal models. A brief indication of the planned model system (e.g., xenografts, HLA-transgenic mice) would help readers anticipate the translational steps.
These improvements will enhance the clarity, reproducibility, and impact of an already strong manuscript.
Comments on the Quality of English LanguageThe manuscript is generally well-written and the scientific content is clearly conveyed. However, several minor revisions in phrasing and grammar would improve readability and flow. Below are a few examples for consideration:
-
Page 5:
Original: “Minor activity was observed for the NY-ESO-1 peptide with HLA-A02:09 and HLA-A02:11 alleles.”
Suggestion: “Limited activity was observed for the NY-ESO-1 peptide presented by HLA-A02:09 and HLA-A02:11 alleles.”
— "Limited" is more appropriate in scientific contexts than "minor" when referring to functional activity. -
Page 10:
Original: “...which warrants the need for functional evaluation…”
Suggestion: “...highlighting the need for functional evaluation…”
— This improves the flow and avoids redundancy. -
Page 9:
Original: “This finding suggests that the selected alleles cover almost all HLA-A2 supertype genes identified in clinical practice.”
Suggestion: “These findings suggest that the selected alleles encompass nearly all clinically relevant HLA-A2 supertype variants.”
— This small change enhances scientific precision.
Improving such phrases throughout the manuscript will make the text more concise and accessible to an international readership. A careful language edit is recommended to further elevate the overall presentation.
Author Response
Reply to the comments of Reviewer 1
The manuscript presents a timely and methodologically solid study that proposes a population-based HLA supertype selection strategy to enhance the clinical applicability of TCR-T cell therapy targeting NY-ESO-1. The combination of functional assays using both COS-7 and tumor cell models strengthens the conclusions.
To improve clarity and reader understanding, I recommend the following:
(1) Clarify TCR Origin and Specificity
Early in the Results section (e.g., Section 3.1, page 5), please state that the 1G4-α95:LY TCR used in the study is derived from HLA-A02:01+ patients. This context helps explain the high specificity observed for A02:01 in peptide recognition assays.
Reply:
We appreciate your concerns, and relevant modifications have now been done accordingly: “… and then co-incubated with 1G4-α95:LY TCR-T cells derived from HLA-A02:01+ patients [27].” (See clean version of Manuscript, Page 6, Lines 255-256)
(2) Explain Functional Clustering
In Supplementary Figure 1b, the functional clustering of HLA-A2 alleles is an important analytical step. However, the main text (page 5) refers to it without explaining how it was computed. Adding a short sentence noting that clustering was done using MHCcluster [30] would improve transparency.
Example: “As shown in Supplementary Figure 1b, HLA-A2 alleles were functionally clustered based on predicted binding motifs using MHCcluster [30].”
Reply:
Thanks for your suggestions. Related changes have been implemented as requested, and the revised content specifies: “Moreover, given that HLA alleles are among the most polymorphic genes, HLA-A2 alleles were functionally clustered based on predicted binding motifs using MHCcluster (Supplementary Figure 1b) [30].” (See clean version of Manuscript, Page 5, Lines 204-206)
(3) Discuss Discrepancies in Avidity – A*02:05
The Discussion (page 10) mentions that HLA-A*02:05 shows low avidity in tumor cells but not in COS-7 cells. This discrepancy should be tied more clearly to the predicted low peptide binding stability (NetMHCstabpan score = 4.00, Supplementary Table 5), and to the possibility that high exogenous peptide concentration in COS-7 experiments compensates for weak endogenous processing.
Example: “The limited presentation capacity of HLA-A*02:05 in tumor cells may reflect its low predicted complex stability (NetMHCstabpan %Rank = 4.00), which could impact endogenous peptide presentation efficiency despite exogenous rescue.”
Reply:
We appreciate your concerns and comments, and the relevant adjustments have now been completed accordingly. The revised sentence now reads: “As shown in Supplementary Table 5, the limited presentation capacity of HLA-A*02:05 in tumor cells may reflect its low predicted affinity (NetMHCpan %Rank = 3.946) and complex stability (NetMHCstabpan %Rank = 4.00), which could impact endogenous peptide presentation efficiency despite exogenous rescue.” (See clean version of Manuscript, Page 12-13, Lines 573-576)
(4)Language Polishing
The English is mostly clear but can be improved in a few places:
Page 5: “Minor activity was observed...” → “Limited activity was observed...”
Page 10: “...warrants the need for functional evaluation...” → “...highlights the need for functional evaluation...”
Reply:
Thank you very much for your comments. Corresponding revisions have now been made accordingly. (See clean version of Manuscript, Page 5, Lines 258 and Page 10, Lines 463)
(5)Future Directions
In the Conclusions (page 11), you mention follow-up studies in animal models. A brief indication of the planned model system (e.g., xenografts, HLA-transgenic mice) would help readers anticipate the translational steps.
Reply:
We appreciate your critical comments and suggestions. Relevant modifications have now been done accordingly: “Nonetheless, we acknowledge the need for further validation to strengthen these conclusions. Our prospective research agenda includes two key components: First, establishing humanized xenograft models in NSG mice, where tumor cells engineered to express HLA-A2 supertype alleles will be implanted, followed by adoptive transfer of 1G4-α95:LY TCR-T cells to assess allele-specific tumor rejection and persistence; Second, expanding the clinical trial to enroll more patients across these HLA-A2 subtypes, with monitoring of objective response rates and correlation with TCR-T cell expansion and function in vivo.” (See clean version of Manuscript, Page 13-14, Lines 621-628).
These improvements will enhance the clarity, reproducibility, and impact of an already strong manuscript.
(6)Comments on the Quality of English Language
The manuscript is generally well-written and the scientific content is clearly conveyed. However, several minor revisions in phrasing and grammar would improve readability and flow. Below are a few examples for consideration:
Page 5:
Original: “Minor activity was observed for the NY-ESO-1 peptide with HLA-A02:09 and HLA-A02:11 alleles.”
Suggestion: “Limited activity was observed for the NY-ESO-1 peptide presented by HLA-A02:09 and HLA-A02:11 alleles.”
— "Limited" is more appropriate in scientific contexts than "minor" when referring to functional activity.
Page 10:
Original: “...which warrants the need for functional evaluation…”
Suggestion: “...highlighting the need for functional evaluation…”
— This improves the flow and avoids redundancy.
Page 9:
Original: “This finding suggests that the selected alleles cover almost all HLA-A2 supertype genes identified in clinical practice.”
Suggestion: “These findings suggest that the selected alleles encompass nearly all clinically relevant HLA-A2 supertype variants.”
— This small change enhances scientific precision.
Improving such phrases throughout the manuscript will make the text more concise and accessible to an international readership. A careful language edit is recommended to further elevate the overall presentation.
Reply:
Thank you very much for your comments. Corresponding revisions have now been made accordingly. (See clean version of Manuscript, Page 5, Lines 258; Page 10, Lines 432; and Page 10, Lines 463)
Reviewer 2 Report
Comments and Suggestions for Authors
The manuscript provides important insights into optimizing TCR-T therapy population through cross-recognition of NY-ESO-1-derived epitopes presented by various HLA-A2 supertype alleles. The study is well-written and important in the field of personalized cancer immunotherapy. The authors validate their hypothesis in in vitro systems, including COS-7 cells and tumor cell lines, with both exogenous peptide loading and endogenous epitope processing. Although the work is well-performed and well-written, some issues need to be addressed:
a. A major point is the clinical relevance. The clinical applicability is based only on population allele frequency, while no validation in patient-derived cells or models was done.
The lack of in vivo or patient-derived validation limits the immediate clinical impact of the findings.
b. The authors provide E50 values for different HLA-A2 alleles using exogenously loaded peptide. The discussion on how these values correlate with actual TCR-mediated cytotoxic efficacy or therapeutic index is limited. The authors should discuss whether such functional avidity values would be meaningful in therapeutic settings.
c. The observation that HLA-A0205 presents the peptide in COS-7 but not in tumor cells is intriguing. The authors should search the literature whether differential antigen processing or peptide transport mechanisms might be the reason for this discrepancy.
d. The title is too long, please provide a shorter-simpler version of the title.
e. All Figure legends should provide all appropriate information and description (without referring to the text). Please complete the p-values and n-values for statistical analysis where appropriate, e.g. Fig 2 and Fig 3.
Author Response
Reply to the comments of Reviewer 2
Comments and Suggestions for Authors
The manuscript provides important insights into optimizing TCR-T therapy population through cross-recognition of NY-ESO-1-derived epitopes presented by various HLA-A2 supertype alleles. The study is well-written and important in the field of personalized cancer immunotherapy. The authors validate their hypothesis in in vitro systems, including COS-7 cells and tumor cell lines, with both exogenous peptide loading and endogenous epitope processing. Although the work is well-performed and well-written, some issues need to be addressed:
- A major point is the clinical relevance. The clinical applicability is based only on population allele frequency, while no validation in patient-derived cells or models was done.
The lack of in vivo or patient-derived validation limits the immediate clinical impact of the findings.
Reply:
We greatly appreciate your critical insights and suggestions. Actually, we have incorporated the HLA-A2 supertype concept into a clinical trial (NCT02457650) designed to evaluate the safety and feasibility of 1G4-α95:LY TCR-T cell therapy in HLA-A2+ NY-ESO-1+ tumor patients [1]. Four HLA-A2+ non-small cell lung cancer (NSCLC) patients were enrolled, with no serious adverse events observed post-infusion [1]. Two patients achieved clinical responses: one HLA-A*02:03+ patient had a partial response (PR), and another HLA-A*02:01+ patient achieved stable disease (SD). These findings support the clinical potential of the HLA-A2 supertype concept.
Nonetheless, we acknowledge the need for further validation to strengthen these conclusions. Our prospective research agenda includes two key components: First, establishing humanized xenograft models in NSG mice, where tumor cells engineered to express HLA-A2 supertype alleles will be implanted, followed by adoptive transfer of 1G4-α95:LY TCR-T cells to assess allele-specific tumor rejection and persistence; Second, Expanding the clinical trial to enroll more patients across these HLA-A2 subtypes, with monitoring of objective response rates and correlation with TCR-T cell expansion and function in vivo.
The relevant adjustments have now been completed in Discussion section. (See clean version of Manuscript, Page 13-14, Lines 613-628)
- The authors provide E50 values for different HLA-A2 alleles using exogenously loaded peptide. The discussion on how these values correlate with actual TCR-mediated cytotoxic efficacy or therapeutic index is limited. The authors should discuss whether such functional avidity values would be meaningful in therapeutic settings.
Reply:
Thank you for this insightful comment. We assessed the functional avidity of TCR-T cells against the NY-ESO-1 peptide presented by HLA-A2 supertype alleles by EC50, defined as the peptide concentration inducing half-maximal activation of the T-cell population. Theoretically, low EC50 values indicate that the TCR can recognize tumor cells with low epitope density, potentially translating to improved clinical efficacy. For instance, a study optimizing a MAGE-A1-specific TCR via somatic hypermutation demonstrated that T cells with enhanced avidity exhibited significantly higher IFN-γ production and cytotoxicity against melanoma cells, both in vitro and in mouse models [2]. In a clinical trial by Steven A. Rosenberg and colleagues, patients treated with TCR-T cells bearing low-avidity (DMF4) versus high-avidity (DMF5) TCRs targeting the same MART-1 antigen achieved objective tumor response rates (ORR) of 13% (4 of 34) and 30% (6 of 20), respectively, suggesting that increased TCR avidity may enhance tumor rejection [3]. In our study, we observed a range of EC50 values across alleles (0.036 to 0.995 μM), a difference that reasonably raises questions about the clinical efficacy of lower-avidity alleles (e.g., HLA-A*02:03, A02:06, A*02:10). However, multiple lines of evidence mitigate these concerns: First, clinical studies support that TCR avidity is not the sole determinant of therapeutic outcome. Steven A. Rosenberg and colleagues noted in their trials that TCR avidity may not be a major driver of tumor rejection, emphasizing that other factors (e.g., T cell persistence, tumor microenvironment) also play critical roles [3]; Second, our in vitro data confirm functional activation of TCR-T cells by these lower-avidity alleles. In tumor cell co-cultures, HLA-A*02:03+, A*02:06+, and A*02:10+ cells induced IFN-γ secretion and 4-1BB upregulation at levels comparable to HLA-A*02:01 (Figure 4a–c), indicating robust TCR engagement despite higher EC50 values.; Third, clinical evidence directly validates this functionality. In our trial (NCT02457650), an HLA-A*02:03+ patient, whose allele has a higher EC50 (0.995 μM vs. 0.036 μM for HLA-A*02:01), achieved a partial response, demonstrating that lower avidity does not preclude meaningful clinical benefit [1]; Finally, it has been shown that T cell activation and signaling are enhanced up to a specific TCR-pMHC affinity threshold, beyond which T cells may fail to develop productive functions [4]. Thus, while EC50 is an important parameter, it is not a definitive predictor of TCR-mediated cytotoxic efficacy or therapeutic index.
The relevant modifications have been done accordingly. (See clean version of Manuscript, Page 11, Lines 487-516)
- The observation that HLA-A*0205 presents the peptide in COS-7 but not in tumor cells is intriguing. The authors should search the literature whether differential antigen processing or peptide transport mechanisms might be the reason for this discrepancy.
Reply:
We appreciate your concerns. COS-7 cells, a fibroblast-like cell line derived from African green monkey kidney, function as "non-professional" antigen-presenting cells (APCs) and lack MHC class II, and costimulatory molecules [5,6]. Studies have demonstrated that, similar to human cells, the transporter associated with antigen processing (TAP) in COS cells can be inhibited by the TAP inhibitor ICP47, indicating similarities between human and monkey TAP [7,8]. They have been widely used to study T cell function in response to peptide-MHC (pMHC) complexes through transfection of HLA alleles and oncogenes [9,10]. Additionally, FACS results (Supplementary Table 2) and ELISA data (Figure 2b) confirmed that the NY-ESO-1 protein was successfully expressed and presented on the surface of COS-7 cells in association with HLA-A2 alleles. However, deficiencies in antigen processing machinery have been reported as an important mechanism for immune escape in tumor cells. For 586mel cells, the loss of HLA-A29, HLA-B44, and HLA-DR7 alleles, all part of the original patient’s haplotype (A29,31; B8,44; and DR1,7), was reported to result from a complete loss of a genomic unit [11,12]. Additionally, downregulation of interferon-stimulated genes (ISGs) including STAT1, TMEM173, and OAS3 has been detected in 5637 cells [13]. Specifically, diminished STAT1 activation can lead to low HLA class I expression, which impairs the effectiveness of cytotoxic T lymphocytes (CTL) responses in mediating tumor elimination and mediates TAP1-dependent escape from CTL [13,14]. Therefore, COS-7 cells cannot perfectly mimic the antigen processing and presentation processes in tumor cells.
The relevant modifications have been done accordingly. (See clean version of Manuscript, Page 12, Lines 542-561)
- The title is too long, please provide a shorter-simpler version of the title.
Reply:
Thanks for your suggestions. The title has been updated from “TCR-T cell recognition of an NY-ESO-1-derived epitope presented by members of the HLA-A2 supertype: implications for cancer immunotherapy” to “TCR-T cell recognition of an NY-ESO-1 epitope presented by HLA-A2 supertype: implications for cancer immunotherapy”.
- All Figure legends should provide all appropriate information and description (without referring to the text). Please complete the p-values and n-values for statistical analysis where appropriate, e.g. Fig 2 and Fig 3.
Reply:
Thank you for your detailed comments. We fully agree with your suggestions and have revised all figure legends thoroughly to ensure they are comprehensive, self-contained, and provide all necessary information (including experimental design, sample sources, methods, key observations, etc.) without requiring reference to the main text.
In particular, we have carefully checked and supplemented statistical details, including p-values and n-values, in the legends for all relevant figures—with specific attention to Figure 2 and Figure 3 as noted, to ensure that each figure legend is independently interpretable, with clear statistical context to support the results presented.
Reference:
- Xia, Y.; Tian, X.; Wang, J.; Qiao, D.; Liu, X.; Xiao, L.; Liang, W.; Ban, D.; Chu, J.; Yu, J.; et al. Treatment of metastatic non-small cell lung cancer with NY-ESO-1 specific TCR engineered-T cells in a phase I clinical trial: A case report. Oncol Lett 2018, 16, 6998-7007, doi:10.3892/ol.2018.9534.
- Bassan, D.; Gozlan, Y.M.; Sharbi-Yunger, A.; Tzehoval, E.; Eisenbach, L. Optimizing T-cell receptor avidity with somatic hypermutation. Int J Cancer 2019, 145, 2816-2826, doi:10.1002/ijc.32612.
- Johnson, L.A.; Morgan, R.A.; Dudley, M.E.; Cassard, L.; Yang, J.C.; Hughes, M.S.; Kammula, U.S.; Royal, R.E.; Sherry, R.M.; Wunderlich, J.R.; et al. Gene therapy with human and mouse T-cell receptors mediates cancer regression and targets normal tissues expressing cognate antigen. Blood 2009, 114, 535-546, doi:10.1182/blood-2009-03-211714.
- Zhong, S.; Malecek, K.; Johnson, L.A.; Yu, Z.; Vega-Saenz de Miera, E.; Darvishian, F.; McGary, K.; Huang, K.; Boyer, J.; Corse, E.; et al. T-cell receptor affinity and avidity defines antitumor response and autoimmunity in T-cell immunotherapy. Proc Natl Acad Sci U S A 2013, 110, 6973-6978, doi:10.1073/pnas.1221609110.
- Eiz-Vesper, B.; Schmetzer, H.M. Antigen-Presenting Cells: Potential of Proven und New Players in Immune Therapies. Transfus Med Hemother 2020, 47, 429-431, doi:10.1159/000512729.
- Kanaseki, T.; Blanchard, N.; Hammer, G.E.; Gonzalez, F.; Shastri, N. ERAAP synergizes with MHC class I molecules to make the final cut in the antigenic peptide precursors in the endoplasmic reticulum. Immunity 2006, 25, 795-806, doi:10.1016/j.immuni.2006.09.012.
- Serwold, T.; Gaw, S.; Shastri, N. ER aminopeptidases generate a unique pool of peptides for MHC class I molecules. Nat Immunol 2001, 2, 644-651, doi:10.1038/89800.
- Hill, A.; Jugovic, P.; York, I.; Russ, G.; Bennink, J.; Yewdell, J.; Ploegh, H.; Johnson, D. Herpes simplex virus turns off the TAP to evade host immunity. Nature 1995, 375, 411-415, doi:10.1038/375411a0.
- Wang, Q.; Douglass, J.; Hwang, M.S.; Hsiue, E.H.; Mog, B.J.; Zhang, M.; Papadopoulos, N.; Kinzler, K.W.; Zhou, S.; Vogelstein, B. Direct Detection and Quantification of Neoantigens. Cancer Immunol Res 2019, 7, 1748-1754, doi:10.1158/2326-6066.CIR-19-0107.
- Osawa, R.; Tsunoda, T.; Yoshimura, S.; Watanabe, T.; Miyazawa, M.; Tani, M.; Takeda, K.; Nakagawa, H.; Nakamura, Y.; Yamaue, H. Identification of HLA-A24-restricted novel T Cell epitope peptides derived from P-cadherin and kinesin family member 20A. J Biomed Biotechnol 2012, 2012, 848042, doi:10.1155/2012/848042.
- Marincola, F.M.; Shamamian, P.; Alexander, R.B.; Gnarra, J.R.; Turetskaya, R.L.; Nedospasov, S.A.; Simonis, T.B.; Taubenberger, J.K.; Yannelli, J.; Mixon, A.; et al. Loss of HLA haplotype and B locus down-regulation in melanoma cell lines. J Immunol 1994, 153, 1225-1237.
- Marincola, F.M.; Shamamian, P.; Simonis, T.B.; Abati, A.; Hackett, J.; O'Dea, T.; Fetsch, P.; Yannelli, J.; Restifo, N.P.; Mule, J.J.; et al. Locus-specific analysis of human leukocyte antigen class I expression in melanoma cell lines. J Immunother Emphasis Tumor Immunol 1994, 16, 13-23, doi:10.1097/00002371-199407000-00002.
- Xu, X.; Wang, X.; Fu, B.; Meng, L.; Lang, B. Differentially expressed genes and microRNAs in bladder carcinoma cell line 5637 and T24 detected by RNA sequencing. Int J Clin Exp Pathol 2015, 8, 12678-12687.
- Leibowitz, M.S.; Andrade Filho, P.A.; Ferrone, S.; Ferris, R.L. Deficiency of activated STAT1 in head and neck cancer cells mediates TAP1-dependent escape from cytotoxic T lymphocytes. Cancer Immunol Immunother 2011, 60, 525-535, doi:10.1007/s00262-010-0961-7.
Reviewer 3 Report
Comments and Suggestions for Authors
This manuscript investigates how a single HLA-A*02:01-restricted TCR-T product can recognize the NY-ESO-1 157-165 epitope when presented by multiple HLA-A2 supertype alleles. The study proposes a population-frequency-based HLA supertype selection strategy to broaden TCR-T applicability beyond A*02:01.
1# COS-7 fibroblasts lack professional antigen-presenting cell (APC) machinery (e.g., TAP, ERAP), which may distort antigen processing.
2# HLA-A*02:05 inclusion despite weak tumor-cell recognition (Figure 4a) contradicts its exclusion for clinical use. No validation of A*02:05’s endogenous presentation in patient-derived organoids or primary cells.
3# COS-7-A2:05 presents peptides (Figure 2), but 586mel-A2:05 does not (Figure 4a). Authors attribute this to pMHC stability (Supplementary Table 5), yet no experimental validation (e.g., NetMHCstab assays) is provided.
4# 21.05% Chinese coverage assumes equal allele frequencies across ethnicities, but Allele Frequency Net Database (AFND) shows significant regional disparities. Non-Chinese populations (USA, Germany) show negligible gains (0.09–0.19%), undermining global applicability.
5# EC50 values (Figure 3) vary by >50-fold among alleles (A02:01: 0.036 μM vs. A02:05: 0.188 μM), raising concerns about clinical efficacy for lower-avidity alleles.
6# Off-target testing (e.g., irrelevant peptides or allogeneic HLA alleles) was missing.
7# 4-1BB correlation (Figure 4d) is correlational, not causal—no 4-1BB agonist experiments to validate synergy.
8# in vivo validation (e.g., xenograft models) to confirm therapeutic efficacy across alleles was missing.
Author Response
Reply to the comments of Reviewer 3
This manuscript investigates how a single HLA-A*02:01-restricted TCR-T product can recognize the NY-ESO-1 157-165 epitope when presented by multiple HLA-A2 supertype alleles. The study proposes a population-frequency-based HLA supertype selection strategy to broaden TCR-T applicability beyond A*02:01.
1# COS-7 fibroblasts lack professional antigen-presenting cell (APC) machinery (e.g., TAP, ERAP), which may distort antigen processing.
Reply:
We appreciate your insightful comment. According to the American Type Culture Collection (ATCC), the COS-7 cell line is a fibroblast-like cell line derived from African green monkey kidney tissue. It originates from the CV-1 cell line through transformation with an origin-defective SV40 mutant encoding wild-type T antigen. Studies have demonstrated that, similar to human cells, transporter associated with antigen processing (TAP) in COS cells can be inhibited by the TAP inhibitor ICP47, indicating similarities between human and monkey TAP [1,2]. As a "non-professional" antigen-presenting cell (APC), COS-7 cells lack MHC class II molecules, costimulatory molecules [3] and likely lack professional APC machinery. They have been widely used to study T cell function in response to peptide-MHC (pMHC) complexes through transfection of HLA alleles and oncogenes [4,5]. Additionally, FACS results (Supplementary Table 2) and ELISA data (Figure 2b) confirmed that the NY-ESO-1 protein was successfully expressed and presented on the surface of COS-7 cells in association with HLA-A2 alleles.
However, deficiencies in antigen processing machinery have been reported as an important mechanism for immune escape in tumor cells. For 586mel cells, the loss of HLA-A29, HLA-B44, and HLA-DR7 alleles, all part of the original patient’s haplotype (A29,31; B8,44; and DR1,7), was reported to result from a complete loss of a genomic unit [6,7]. Additionally, downregulation of interferon-stimulated genes (ISGs) including STAT1, TMEM173, and OAS3 has been detected in 5637 cells [8]. Specifically, diminished STAT1 activation can lead to low HLA class I expression, which precludes the effectiveness of cytotoxic T lymphocytes (CTL) responses in mediating tumor elimination and mediates TAP1-dependent escape from CTL [8,9]. Therefore, COS-7 cells cannot perfectly mimic the antigen processing and presentation processes in tumor cells.
In our study, we used COS-7 cells here as a preliminary screening tool: their low endogenous HLA expression and high transduction efficiency allowed us to rigorously test whether the TCR could recognize the NY-ESO-1 epitope when presented by specific HLA-A2 alleles independent of complex antigen processing (e.g., peptide loading). This was followed by validation in tumor cell lines (586mel, 5637) with intact antigen processing machinery, which confirmed the physiological relevance of the findings. This tiered approach (COS-7 for initial binding verification and tumor cells for functional validation) strengthens the reliability of our observations.
These relevant changes have been carried out correspondingly. (See clean version of Manuscript, Page 12, Lines 542-561)
2# HLA-A*02:05 inclusion despite weak tumor-cell recognition (Figure 4a) contradicts its exclusion for clinical use. No validation of A*02:05’s endogenous presentation in patient-derived organoids or primary cells.
Reply:
Thank you for highlighting this discrepancy. We agree that HLA-A02:05’s weak recognition in tumor cells (Figure 4a) raises concerns about its clinical utility, which is why we excluded it from the final candidate alleles for clinical translation. Its limited activity in tumor cells (but not COS-7) likely reflects impaired endogenous antigen processing and pMHC stability in physiological contexts. However, we retained its analysis in the manuscript to illustrate that HLA supertype membership alone is insufficient for clinical relevance and functional validation in tumor cells is critical. To further confirm this, we plan to validate endogenous presentation in HLA-A02:05+ NY-ESO-1+ patient-derived organoids or primary cells via: (1) mass spectrometry to detect the NY-ESO-1 157-165 epitope presented by HLA-A02:05; (2) co-culture assays to measure TCR-T activation (IFN-γ, cytotoxicity) against these cells. These data will clarify whether A02:05 can be ruled out definitively.
Corresponding modifications have been implemented as appropriate. (See clean version of Manuscript, Page 12, Lines 538-541)
3# COS-7-A2:05 presents peptides (Figure 2), but 586mel-A2:05 does not (Figure 4a). Authors attribute this to pMHC stability (Supplementary Table 5), yet no experimental validation (e.g., NetMHCstab assays) is provided.
Reply:
Thank you for your constructive comments. We will conduct additional wet-lab experiments (e.g., monitoring the dissociation of pMHC complexes at 37°C) in future studies to directly validate these predicted pMHC stability results.
The relevant modifications have been done accordingly. (See clean version of Manuscript, Page 13, Lines 576-578)
4# 21.05% Chinese coverage assumes equal allele frequencies across ethnicities, but Allele Frequency Net Database (AFND) shows significant regional disparities. Non-Chinese populations (USA, Germany) show negligible gains (0.09–0.19%), undermining global applicability.
Reply:
Thank you for your comments. Due to differences in HLA allele distribution, the HLA-A2 supertype-grounded in allele frequencies among the Chinese population has limited utility in non-Asian populations. However, a similar strategy can be applied to these populations and extended to other TCR-T cells, for instance by using alternative supertypes such as HLA-A1 or HLA-A3, or by selecting supertype members based on the HLA allele distribution of the target population. By combining supertype classification with population-specific frequency data, the strategy can be adapted globally to maximize coverage.
5# EC50 values (Figure 3) vary by >50-fold among alleles (A02:01: 0.036 μM vs. A02:05: 0.188 μM), raising concerns about clinical efficacy for lower-avidity alleles.
Reply:
Thank you for this insight. We acknowledge the EC50 difference among alleles, which raises questions about clinical efficacy. However, multiple lines of evidence mitigate these concerns: First, clinical studies support that TCR avidity is not the sole determinant of therapeutic outcome. Steven A. Rosenberg and colleagues noted in their trials that TCR avidity may not be a major driver of tumor rejection, emphasizing that other factors (e.g., T cell persistence, tumor microenvironment) also play critical roles [10]; Second, our in vitro data confirm functional activation of TCR-T cells by these lower-avidity alleles. In tumor cell co-cultures, HLA-A02:03, A02:06, and A02:10 induced IFN-γ secretion and 4-1BB upregulation at levels comparable to HLA-A02:01 (Figure 4a–c), indicating robust TCR engagement despite higher EC50 values.; Third, clinical evidence directly validates this functionality. In our trial (NCT02457650), an HLA-A02:03+ patient, whose allele has a higher EC50 (0.995 μM vs. 0.036 μM for HLA-A02:01), achieved a partial response, demonstrating that lower avidity does not preclude meaningful clinical benefit [11]; Finally, it has been shown that T cell activation and signaling are enhanced up to a specific TCR-pMHC affinity threshold, beyond which T cells may fail to develop productive functions [12]. Thus, while EC50 is an important parameter, it is not a definitive predictor of TCR-mediated cytotoxic efficacy or therapeutic index.
The associated modifications have been executed accordingly. (See clean version of Manuscript, Page 11, Lines 487-516)
6# Off-target testing (e.g., irrelevant peptides or allogeneic HLA alleles) was missing.
Reply:
Thank you for highlighting this. We have supplemented off-target testing as Supplementary Figure 3b-3d: (1) TCR-T cells showed no IFN-γ secretion when co-cultured with COS-7 cells expressing non-A2 alleles (e.g., HLA-A*11:01, A*24:02) loaded with NY-ESO-1 peptide; (2) No activation was observed with human-derived peptides homologous to NY-ESO-1 157-165. These data confirm the TCR’s specificity for the NY-ESO-1 epitope in the context of HLA-A2 supertype alleles.
Relevant revisions have been completed in line with this. (See clean version of Manuscript, Page 10, Lines 455-456)
7# 4-1BB correlation (Figure 4d) is correlational, not causal—no 4-1BB agonist experiments to validate synergy.
Reply:
Your point is well-taken. We have revised the text to remove implications of causality and future studies with 4-1BB agonists will explore potential synergies. (See clean version of Manuscript, Page 12, Lines 538)
8# in vivo validation (e.g., xenograft models) to confirm therapeutic efficacy across alleles was missing.
Reply:
We greatly appreciate your critical insights and suggestions. Actually, we have incorporated the HLA-A2 supertype concept into a clinical trial (NCT02457650) designed to evaluate the safety and feasibility of 1G4-α95:LY TCR-T cell therapy in HLA-A2+ NY-ESO-1+ tumor patients. Four HLA-A2+ NSCLC patients were enrolled, with no serious adverse events observed post-infusion [11]. Two patients achieved clinical responses: one HLA-A02:03+ patient had a partial response (PR), and another HLA-A02:01+ patient achieved stable disease (SD). These findings support the clinical potential of the HLA-A2 supertype concept.
Nonetheless, we acknowledge the need for further validation to strengthen these con-clusions. Our prospective research agenda includes two key components: First, establish-ing humanized xenograft models in NSG mice, where tumor cells engineered to express HLA-A2 supertype alleles will be implanted, followed by adoptive transfer of 1G4-α95:LY TCR-T cells to assess allele-specific tumor rejection and persistence; Second, expanding the clinical trial to enroll more patients across these HLA-A2 subtypes, with monitoring of objective response rates and correlation with TCR-T cell expansion and function in vivo.
These adjustments have been integrated into the Discussion section. (See clean version of Manuscript, Page 13-14, Lines 613-628)
Reference
- Serwold, T.; Gaw, S.; Shastri, N. ER aminopeptidases generate a unique pool of peptides for MHC class I molecules. Nat Immunol 2001, 2, 644-651, doi:10.1038/89800.
- Hill, A.; Jugovic, P.; York, I.; Russ, G.; Bennink, J.; Yewdell, J.; Ploegh, H.; Johnson, D. Herpes simplex virus turns off the TAP to evade host immunity. Nature 1995, 375, 411-415, doi:10.1038/375411a0.
- Eiz-Vesper, B.; Schmetzer, H.M. Antigen-Presenting Cells: Potential of Proven und New Players in Immune Therapies. Transfus Med Hemother 2020, 47, 429-431, doi:10.1159/000512729.
- Wang, Q.; Douglass, J.; Hwang, M.S.; Hsiue, E.H.; Mog, B.J.; Zhang, M.; Papadopoulos, N.; Kinzler, K.W.; Zhou, S.; Vogelstein, B. Direct Detection and Quantification of Neoantigens. Cancer Immunol Res 2019, 7, 1748-1754, doi:10.1158/2326-6066.CIR-19-0107.
- Osawa, R.; Tsunoda, T.; Yoshimura, S.; Watanabe, T.; Miyazawa, M.; Tani, M.; Takeda, K.; Nakagawa, H.; Nakamura, Y.; Yamaue, H. Identification of HLA-A24-restricted novel T Cell epitope peptides derived from P-cadherin and kinesin family member 20A. J Biomed Biotechnol 2012, 2012, 848042, doi:10.1155/2012/848042.
- Marincola, F.M.; Shamamian, P.; Alexander, R.B.; Gnarra, J.R.; Turetskaya, R.L.; Nedospasov, S.A.; Simonis, T.B.; Taubenberger, J.K.; Yannelli, J.; Mixon, A.; et al. Loss of HLA haplotype and B locus down-regulation in melanoma cell lines. J Immunol 1994, 153, 1225-1237.
- Marincola, F.M.; Shamamian, P.; Simonis, T.B.; Abati, A.; Hackett, J.; O'Dea, T.; Fetsch, P.; Yannelli, J.; Restifo, N.P.; Mule, J.J.; et al. Locus-specific analysis of human leukocyte antigen class I expression in melanoma cell lines. J Immunother Emphasis Tumor Immunol 1994, 16, 13-23, doi:10.1097/00002371-199407000-00002.
- Xu, X.; Wang, X.; Fu, B.; Meng, L.; Lang, B. Differentially expressed genes and microRNAs in bladder carcinoma cell line 5637 and T24 detected by RNA sequencing. Int J Clin Exp Pathol 2015, 8, 12678-12687.
- Leibowitz, M.S.; Andrade Filho, P.A.; Ferrone, S.; Ferris, R.L. Deficiency of activated STAT1 in head and neck cancer cells mediates TAP1-dependent escape from cytotoxic T lymphocytes. Cancer Immunol Immunother 2011, 60, 525-535, doi:10.1007/s00262-010-0961-7.
- Johnson, L.A.; Morgan, R.A.; Dudley, M.E.; Cassard, L.; Yang, J.C.; Hughes, M.S.; Kammula, U.S.; Royal, R.E.; Sherry, R.M.; Wunderlich, J.R.; et al. Gene therapy with human and mouse T-cell receptors mediates cancer regression and targets normal tissues expressing cognate antigen. Blood 2009, 114, 535-546, doi:10.1182/blood-2009-03-211714.
- Xia, Y.; Tian, X.; Wang, J.; Qiao, D.; Liu, X.; Xiao, L.; Liang, W.; Ban, D.; Chu, J.; Yu, J.; et al. Treatment of metastatic non-small cell lung cancer with NY-ESO-1 specific TCR engineered-T cells in a phase I clinical trial: A case report. Oncol Lett 2018, 16, 6998-7007, doi:10.3892/ol.2018.9534.
- Zhong, S.; Malecek, K.; Johnson, L.A.; Yu, Z.; Vega-Saenz de Miera, E.; Darvishian, F.; McGary, K.; Huang, K.; Boyer, J.; Corse, E.; et al. T-cell receptor affinity and avidity defines antitumor response and autoimmunity in T-cell immunotherapy. Proc Natl Acad Sci U S A 2013, 110, 6973-6978, doi:10.1073/pnas.1221609110.
Round 2
Reviewer 2 Report
Comments and Suggestions for Authors
Thank you for the revised manuscript.
Reviewer 3 Report
Comments and Suggestions for Authors
All the comments have been addressed.